# Enhancing the Impact of Individualized Nutrition Therapy with Real-Time Continuous Glucose Monitoring Feedback in Overweight and Obese Individuals with Prediabetes

**DOI:** 10.3390/nu16234005

**Published:** 2024-11-22

**Authors:** Raedeh Basiri, Lawrence J. Cheskin

**Affiliations:** 1Department of Nutrition and Food Studies, George Mason University, Fairfax, VA 22030, USA; lcheskin@gmu.edu; 2Institute for Biohealth Innovation, George Mason University, Fairfax, VA 22030, USA; 3Department of Medicine (GI), Johns Hopkins School of Medicine, Baltimore, MD 21205, USA

**Keywords:** individualized nutrition therapy, personalized nutrition, prediabetes, diabetes, CGM, overweight, obesity, blood glucose control, time in range, glucose CV%, nutrition education

## Abstract

Background/Objectives: prediabetes is a significant risk factor for the development of type 2 diabetes, cardiovascular diseases, chronic kidney disease, and other complications. Early diagnosis of prediabetes, coupled with education on lifestyle changes that support blood glucose management, are crucial for the prevention or delay of type 2 diabetes and related complications. This study aimed to evaluate the impact of incorporating real-time feedback from continuous glucose monitoring (CGM) into individualized nutrition therapy (INT) on blood glucose control in individuals with prediabetes who are overweight or obese. Methods: participants (mean age ± SD: 55 ± 6 years; BMI: 31.1 ± 4.1 kg/m²) were randomly assigned to either the treatment group (*n* = 15) or the control group (*n* = 15). Both groups received INT and CGM, but the control group was blinded to the CGM data until the end of this study. Participants were followed for 30 days and visited the lab every 10 days for CGM replacement, study measurements, and dietary consultations. Results: the treatment group showed a significant increase in the percentage of time spent in the target blood glucose range (*p* = 0.02) and a significant decrease in the mean blood glucose concentration (*p* < 0.05), glucose management indicator (*p* = 0.02), percent coefficient of variation for blood glucose (*p* = 0.01), and percent time spent in the high or very high blood glucose ranges (*p* = 0.04). These changes were not statistically significant for the control group. Conclusions: adding CGM feedback to INT resulted in better management of blood glucose levels in overweight or obese individuals with prediabetes.

## 1. Introduction

Prediabetes represents a significant medical and public health concern and is characterized by impaired fasting glucose and/or impaired glucose tolerance [1] but not to the degree that is diagnostic of type 2 diabetes. Statistics reveal that over 98 million American adults, more than one-third of the population, are affected by prediabetes [2]. Prediabetes predisposes individuals to an elevated risk of developing type 2 diabetes, as well as cardiovascular complications such as heart disease and stroke [2]. Notably, over 80% of individuals with prediabetes are unaware of their condition [2], which could put them at greater risk of progression and related complications. The annual rate of progression from prediabetes to type 2 diabetes is estimated to be around 5–10% [3]. The remaining lifetime risk of progressing from prediabetes to overt diabetes at age 45 is reported to be as high as 57.5% for individuals meeting the American Diabetes Association (ADA) criteria for prediabetes [4]. Early identification and management of prediabetes are crucial to reduce the risk of progressing to type 2 diabetes and associated complications. Globally, 537 million adults aged 20 to 79—about 1 in 10—are currently living with diabetes [5]. This number is expected to increase significantly, reaching 643 million by 2030 and 783 million by 2045. In 2021 alone, diabetes was responsible for 6.7 million deaths, equating to 1 death every five seconds [5]. The financial burden is also substantial, with diabetes-related health expenditure reaching at least USD 966 billion, a 316% increase over the past 15 years [5]. Additionally, 541 million adults have impaired glucose tolerance (IGT), putting them at a high risk of developing type 2 diabetes. Lifestyle changes, particularly those focused on diet and physical activity, are among the most effective interventions for better blood glucose control and prevention of related complications [6,7,8,9,10]; however, overly restrictive dietary recommendations are often not sustainable in the long term. It has been reported that only a small fraction of patients with diabetes follow dietary recommendations by a dietitian [11,12]. Factors such as calorie restrictions, lack of dietary education, and inability to afford a healthful diet are important factors affecting adherence to dietary recommendations [13,14,15,16].

Currently, the main focus of dietary interventions for individuals who are overweight or obese and are at risk of chronic diseases such as diabetes is weight loss, typically achieved through low-calorie and low-carbohydrate diets. Although these approaches have been proven to be effective in the short term [17], they are often not sustainable due to various limitations such as stress, physical discomfort, emotional problems, and the psychological effects of dieting, including depression and anxiety [18,19,20].

Studies using continuous glucose monitoring (CGM) in healthy individuals have shown significant inter-individual variability in postprandial blood glucose responses to the same foods [21,22]. Given the heterogeneity of metabolic responses among individuals, recommendations for dietary modifications should be designed based on each individual’s biological needs and personal preferences to lessen the burden of significant changes in nutritional habits. Personalized dietary interventions coupled with real-time feedback from CGM can enhance adherence and long-term success in managing blood glucose levels and reducing the risk of progressing to type 2 diabetes in at-risk populations. Integrating real-time feedback from CGM can be pivotal in maximizing the impact of dietary interventions by showing individuals how foods influence their blood glucose levels, empowering them with actionable insights and consequently reducing anxiety. It has been shown that positive psychological reactions, including reduced stress and anxiety, greater peace of mind, and a greater sense of normalcy, can be achieved through effective CGM use [23]. A randomized controlled trial by Anh et al. showed that noncontact dietary coaching, combined with CGM, improved behavioral skills and health outcomes in adults with prediabetes or diabetes [24]. Moreover, it has been shown that a personalized dietary intervention approach combined with CGM can promote better adherence to dietary recommendations [25].

This study aimed to examine the effects of adding real-time feedback from CGM to individualized nutrition therapy (INT) on indices of blood glucose control in individuals with prediabetes who were overweight or obese.

## 2. Materials and Methods

### 2.1. Study Design

This randomized clinical trial evaluated the clinical efficacy of incorporating real-time feedback from CGM into INT for blood glucose management in individuals with prediabetes who were overweight or obese. This study was registered on ClinicalTrials.gov (NCT05161897) and the protocol was approved by the George Mason University Institutional Review Board (IRB). All participants completed written informed consent before the start of this study. Inclusion criteria were nonpregnant/nonlactating individuals between ages 45 and 65 years from any race or ethnicity who had a baseline hemoglobin A1c (HbA1c) of 5.7% to 6.4% (prediabetes) [26] and a body mass index (BMI) between 25 and 39.9 kg/m^2^. The rationale for the age range is that the prevalence of prediabetes is higher in this age group [27,28]. Exclusions were active cancer; thyroid, kidney, liver, and pancreatic diseases; heavy cigarette smokers (≥25 cigarettes a day); consumers of more than 12 alcoholic drinks/week on average; those with major dietary restrictions that could potentially limit the ability to deliver effective dietary interventions; participating in any weight loss or dietary program/taking prescribed appetite suppressants; or participating in another investigational study concurrently.

#### 2.1.1. Prescreening Telephone Interview

Individuals who inquired about this study were given a brief overview of the study and, if interested, prescreened by telephone to determine eligibility before scheduling their informed consent and in-person visits.

#### 2.1.2. Screening/Baseline

During the screening visit, participants’ HbA1c and anthropometric measurements were obtained. If they were eligible for this study, they were asked to complete a medical and medication history questionnaire. Participants were then randomized into treatment (*n* = 15) or control groups (*n* = 15) using an online random number generator. All participants were asked to continue with their typical dietary intake during the first 10 days of this study. Both groups were also instructed to record their dietary intake throughout this study using a food diary form. Participants were provided with a CGM to be used for assessing continuous glucose concentrations and glycemic variability throughout the study. They were then scheduled for a second clinic visit 10 days later to complete study measurements, replace their CGM, and receive individualized dietary recommendations from a dietitian. During this visit, all participants were given dietary recommendations tailored to their energy requirements for weight maintenance using the Mifflin St. Jeor formula [29], with a recommended macronutrient distribution of 50% carbohydrates, 20% protein, and 30% fat, constituting a moderate carbohydrate diet [30]. The rationale for choosing this diet was that the aforementioned percentage of macronutrients has been shown to be effective in improving blood glucose and cardiovascular disease risk factors [30]. Both groups were also provided with guidance on the distribution of carbohydrate servings throughout the day and were educated about preferred carbohydrate choices [31]. No recommendations were made related to a change in physical activity. All participants received a new CGM and dietary recommendations and were given an opportunity to seek clarification and ask questions about their prescribed diets. Additionally, the treatment group had real-time access to their blood glucose levels via their cell phones, receiving alerts if their glucose concentrations rose above 140 mg/dL or dropped below 70 mg/dL. They also had the opportunity to review their food diary with the dietitian and compare it with the recorded data from the CGM. While reviewing CGM recorded data, foods that resulted in a blood glucose measure of more than 140 mg/dL were flagged as undesirable foods that needed to be consumed less frequently or in a lower amount. In contrast, both the dietitian and control group participants were blinded to the CGM recordings until the end of this study; thus, the CGM data were not included in nutrition education for the control group. At visits three and four, participants were also asked how well they felt they were able to follow the provided recommendations. If participants reported any obstacles in adherence, the dietitian collaborated with them to address these issues and identify practical alternatives, enhancing the likelihood of successful adherence to the recommendations. All participants were followed for 30 days, with visits every 10 days for CGM replacement and study measurements. During these visits, both groups also had the opportunity to ask questions and receive dietary recommendations from the dietitian.

### 2.2. Study Measurements

#### 2.2.1. Finger Stick Blood (at Screening)

To confirm the eligibility of participants, finger stick blood collection was performed at the screening visit to measure HbA1C at the point of care using an Abbott AFINION 2 Portable Analyzer (Abbott Laboratories, Chicago, IL, USA).

#### 2.2.2. Anthropometric Measurements (at Baseline and Every 10 Days Until the End of the Study)

Height without shoes was measured using a wall-mounted stadiometer, weight was measured using a digital scale (Health o meter^®^ Professional Scales, McCook, IL, USA), and BMI was calculated by the formula BMI = weight (kg)/[height (m)^2^].

#### 2.2.3. Food Diary (at Baseline and Every 10 Days Until the End of the Study)

All participants were provided with a food diary form and were asked to record all foods and beverages consumed during each day of this study. They were instructed to include detailed information about the name and amount of ingredients for each food item, the time of consumption, and the method of preparation. These data were then used by the dietitian to relate the foods to the blood glucose recorded on the CGM to flag the potential foods that caused a spike in blood glucose.

#### 2.2.4. Continuous Glucose Monitoring (at Baseline and Every 10 Days Until the End of the Study)

A CGM device was inserted into the periumbilical region of the abdomen, following the manufacturer’s instructions, to collect and assess 24-hour glucose concentrations and variability during this study. Participants wore the device for 10 days and then returned to the clinic for CGM replacement. Data recorded on the CGM were downloaded using a CGM reader, and the following measurements were used for analysis:Glucose management indicator (GMI).

GMI approximates the laboratory HbA1c level expected based on average blood glucose levels measured using CGM data. The formula [32] to calculate GMI is
GMI (%) = 3.31 + 0.02392 × [mean glucose in mg/dL]
Percent coefficient of variation (%CV).

Glycemic variability is considered a contributing factor to the risk of long-term diabetes-related complications, and in the short term, it is associated with episodes of hypoglycemia and hyperglycemia [33,34]. The %CV is calculated as the standard deviation (SD) of measured glucose values observed during the use of CGM divided by the mean of measured glucose values in the same observation period multiplied by 100 [35].
Percentage of time spent in very high, high, and target blood glucose ranges: time in range (TIR).

Since our participants had prediabetes, we assessed the effects of our intervention on blood glucose concentration using two different approaches: one based on the recommendations for patients with diabetes and another based on blood glucose ranges for individuals with normoglycemia.

In our first model, we used recommendations for blood glucose control for patients with diabetes from the American Diabetes Association [36]; therefore, blood glucose concentrations above 250 mg/dL were defined as the very high range, 180 to 250 mg/dL as high, and 70 to 180 mg/dL as TIR. In the second model, blood glucose concentrations above 200 mg/dL were defined as the very high range, 140 to 200 mg/dL as high, and 70 to 140 mg/dL as TIR.

### 2.3. Statistical Analysis

A power analysis was performed to determine the required sample size for detecting a statistically significant interaction between the intervention (INT versus control) and time (across four time points). The calculation was based on an effect size of 0.25, representing the mean difference and standard deviation for the primary outcome of interest: a clinically meaningful reduction in glucose variability over 30 days [37]. Using G*Power software (version 3.1.9.4), we estimated that a sample of at least 24 participants (12 per group) would be needed to achieve 80% power at a 0.05 significance level. To allow for an anticipated 20% dropout or missing data rate, we ultimately enrolled 30 participants.

Data analysis was performed using Statistical Package for the Social Sciences (SPSS) version 29.0 (SPSS, Inc., Chicago, IL, USA), with a significance threshold set at *p* < 0.05 for all tests. Descriptive statistics were performed to evaluate population characteristics, and an analysis of variance (ANOVA) table was used to assess the distribution of covariates between the groups. If there was a significant difference in the distribution of a potential confounding variable between the groups, the variable was included as a covariate in the model to minimize its potential effects on outcome measures. A general linear model with repeated measures test was used to evaluate changes in indices of blood glucose control throughout the study.

## 3. Results

At baseline, the mean± SD for age was 55 ± 6 years, and for body mass index (BMI) it was 31.1 ± 4.1 kg/m^2^. Except for one participant in the control group who had been taking metformin 500 mg/day for over three months, the participants were not using any blood glucose-lowering medications. The distribution of potential covariates was evaluated, and no significant differences were observed for sex, ethnicity, BMI, and hemoglobin A1c (HbA1c) between groups; however, the average age of participants was significantly higher in the treatment group compared to the control group. Therefore, age was included as a covariate in the models. Given the established relationship between BMI and blood glucose control, BMI was also included as a covariate in the models to account for its potential confounding effects. The baseline characteristics of participants in each group are reported in Table 1.

### 3.1. Average Blood Glucose Levels

In the treatment group, the average blood glucose significantly decreased from 129.1 ± 4.3 to 121.6 ± 4.9 mg/dL (*p* < 0.05). The average blood glucose went from 131.1 ± 4.7 to 129.5 ± 5.3 mg/dL in the control group; however, these changes were not statistically significant.

### 3.2. Glucose Management Indicator (GMI)

The treatment group significantly lowered their GMI from 6.4% to 6.2% during the study (*p* = 0.02); however, the average GMI remained the same (6.4%) in the control group.

### 3.3. Percent Coefficient of Variation (%CV) for Blood Glucose Levels

A consistent and significant reduction was observed in %CV from the baseline (mean change= 2.6%, *p*= 0.01) for the treatment group; however, the changes in %CV were not statistically significant for the control group. The mean changes in the %CV for both groups are depicted in Figure 1.

### 3.4. Percentage of Time Spent in the Very High Blood Glucose Ranges

Based on the results of the first model, which defined blood glucose concentrations above 250 mg/dL as very high, the treatment group consistently reduced the percentage of time spent in the very high blood glucose range from 0.27% to 0.18%. In contrast, in the control group, the percentage of time spent in the very high blood glucose range decreased from 0.47% at visit one to 0.20% at visit two but then increased to 0.55% at visit three. There were no statistically significant changes in the percentage of time spent in the very high blood glucose range in either group. However, the reduction was consistent in the treatment group, while the control group showed variability during this study. Figure 2 illustrates the changes in the percentage of time spent with blood glucose levels exceeding 250 mg/dL for both groups.

In the second model, in which the blood glucose concentrations >200 mg/dL were defined as very high, the average time spent in the very high range significantly decreased from 0.9% to 0.08% (*p* = 0.04) in the treatment group. In contrast, in the control group, the percentage of time spent in the very high range decreased from 2.7 mg/dL to 1.3 mg/dL between visits one and two, but this percentage subsequently increased to 2.9% by the third visit. Figure 3 presents the changes in the average percentage of time that blood glucose levels exceeded 200 mg/dL in both groups.

### 3.5. Percentage of Time Spent in the High Blood Glucose Ranges

In the first model, where blood glucose levels between 180 and 250 mg/dL were defined as a high blood glucose range, a consistent and significant (*p* = 0.005) decrease from 4.4% to 1.3% was observed in the treatment group. In the control group, the percentage of time spent in the high blood glucose range initially decreased from 5.8% to 3.9% between the first and second visits but then rose to 6.4% by the third visit; none of these changes were statistically significant. The mean changes in the percentage of time spent in the high blood glucose range for both groups are shown in Figure 4.

In the second model, where high blood glucose levels were defined as concentrations between 140 and 200 mg/dL, the treatment group demonstrated a significant reduction in the percentage of time spent in this high glucose range from visit one to visit two (from 27.5% to 14.37%; *p* = 0.004). However, there was a slight, nonsignificant increase to 18.1% from visit two to visit three. Conversely, the control group also showed a decrease in the percentage of time spent in the high blood glucose range from visit one (24.8%) to visit two (20.6%) but experienced a subsequent increase to 22.1% from visit two to visit three. Despite a slight increase in the percentage of time spent in high blood glucose range from visits two to three in the treatment group, the overall interaction between time and intervention remained significant for the treatment group (*p* = 0.04) during this study, whereas it was not significant for the control group. Figure 5 depicts the mean changes in the percentage of time spent in the high blood glucose range for the treatment and control groups.

### 3.6. Percentage of Time Spent in Target Blood Glucose Ranges: Time in Range (TIR)

In our first model, the time in range was defined as the percentage of time that the blood glucose concentrations remained between 70 and 180 mg/dL. Participants in the treatment group showed a significant increase in the TIR from 95.1% to 97.9% (*p* = 0.02), but no significant changes were observed in the TIR (93.0%) for the control group. Figure 6 shows the mean changes in the time spent in the TIR for both groups based on Model 1.

When the time spent in the blood glucose concentrations between 70 and 140 mg/dL was considered for TIR, in the second model, the treatment group tended to significantly increase their TIR from 70.8% to 80.1% (*p*= 0.05); however, no significant changes were observed in the control group (70.4% to 72.8%). The mean changes in time spent within the target blood glucose range, as defined by Model 2, are shown in Figure 7.

### 3.7. Changes in Percentage of Time Spent in Low Blood Glucose Range

The blood glucose concentration between 54 and 70 mg/dL was considered a low blood glucose concentration. No significant changes in the average time spent in the low blood glucose range were observed for the treatment or the control group.

### 3.8. Changes in Percentage of Time Spent in Very Low Blood Glucose Range

The concentration of blood glucose of less than 54 mg/dL was considered as very low blood glucose. Only five participants in the treatment group and six in the control group experienced very low blood glucose during this study. No significant changes in the percentage of time spent in the very low blood glucose range were observed in the treatment or the control group. 

### 3.9. Changes in Calorie and Percent Calorie Intake from Macronutrients

During this study, the treatment group reduced their calorie intake from 1774 kcal to 1688 kcal, while the control group decreased from 2148 kcal to 2092 kcal. The percentage of calorie intake from carbohydrates declined from approximately 40% to 36% in the treatment group and from 36% to 34% in the control group. The caloric intake from fat increased from around 37% to 39% in the treatment group and from 39% to 40% in the control group. Similarly, the percentage of calories from protein rose from about 20% to 22% in the treatment group and from 20% to 21% in the control group. None of these changes reached statistical significance, and all values were adjusted for age.

## 4. Discussion

We found that adding real-time feedback from CGM to INT enhanced the effects of dietary intervention on blood glucose management in individuals with prediabetes who were overweight or obese. These results are aligned with a study by Ben-Yacov [38], which indicated that using a personalized postprandial-targeting diet combined with CGM improved glycemic control more significantly than the Mediterranean diet, measured as time spent at blood glucose levels above 140 mg/dL and HbA1c.

Utilizing CGMs to assess the effectiveness of our dietary counseling provided us with important information about GMI, CV, TIR, and the percent time spent in high, very high, low, and very low ranges of blood glucose, important factors to consider when seeking to optimize blood glucose control. It was found that TIR is a strong predictive measure for long-term complications of diabetes, particularly microvascular complications [39]. Furthermore, reduced time at high and very high blood glucose levels in patients with diabetes is related to reduced albuminuria, severity of diabetic retinopathy, and prevalence of peripheral neuropathy and cardiac autonomic neuropathy [40]. Currently, there are no recommendations for TIR for individuals with prediabetes; however, suggested TIR for patients with diabetes is to keep the blood glucose in the range of 70 to 180 mg/dL for >70% of the day, below 70 mg/dL for <4%, and below 54 mg/dL for <1% of the day and to minimize time above 180 mg/dL [41,42]. For individuals without diabetes, normal blood glucose levels after a meal should peak below 140 mg/dL [43]. In this study, we employed two different models, one based on recommendations for patients with diabetes and the other on normal blood glucose ranges, to evaluate the effects of our intervention more comprehensively. In both models, significant improvements were observed in the average blood glucose levels and TIR in the treatment but not in the control group. Although the reduction in time spent in the very high blood glucose range (>250 mg/dL) did not reach statistical significance in either group in Model 1, the treatment group consistently showed a decrease throughout this study. In contrast, the control group exhibited a numerical reduction from visit one to visit two, but this measure increased from visit two to visit three. It is important to note that changes in very high blood glucose levels are clinically significant, even in the absence of statistical significance, given the known adverse effects of levels exceeding 250 mg/dL on the body [36,44,45]. The changes in the high blood glucose followed a similar trend in Model 1, with a significant >3% decrease in the treatment group (*p* = 0.005) but no significant decrease from visit one to two and an increase from visit two to three in the control group. These results suggest that using CGM concurrent with dietary intervention could help with the sustainability of the intervention.

In Model 2, the control group followed a trend similar to that found using Model 1 for changes in time spent in the high blood glucose range and TIR. However, the treatment group experienced a slight increase in time spent in high blood glucose (>140 mg/dL) and a decrease in TIR (blood glucose between 140 and 180 mg/dL) from visit two to visit three. This may be attributed to the short duration of this study, as several participants expressed interest in testing the effects of foods not typically part of their diet, knowing they would not have the opportunity after the final session. When analyzing the percentage of calorie intake from carbohydrates for the treatment group, no significant changes were observed from visit one (39.68 ± 2.7%) to visit two (39.13 ± 2.38%). However, a notable reduction was seen from visit two to visit three (36.43 ± 2.52%), confirming the positive effects of the intervention (INT) when combined with real-time feedback from CGM. Interestingly, no significant changes occurred between visit three and visit four (36.17 ± 2.51%), which aligns with anecdotal reports from participants who admitted to trying new food options during this period, anticipating the end of CGM access. In contrast, the control group showed a different pattern. Their carbohydrate intake decreased from visit one (35.91 ± 2.67%) to visit two (33.79 ± 2.38%) but remained relatively stable during visits three (33.51 ± 2.52%) and four (33.98 ± 2.51%). All reported values are adjusted for age. Longer-term studies may help address these effects by providing participants with sufficient time to adapt to dietary changes. Such an approach would allow participants to explore occasional foods of interest while maintaining more informed and sustainable dietary habits over time.

It is well documented that microvascular complications like retinopathy, neuropathy, and nephropathy, as well as macrovascular complications like cardiovascular disease and kidney disease, can manifest during the prediabetes stage before progression to overt diabetes [46,47,48]. Therefore, the observed reductions in time spent in the very high and high blood glucose ranges and significant improvements in TIR, CV, and blood glucose concentrations in the treatment group suggest a significant potential for preventing the onset of related complications in this population. Longer-duration clinical trials should be conducted to assess the sustainability of similar interventions.

Fluctuations in blood glucose levels are an important risk factor for the development and progression of various microvascular (nephropathy, retinopathy, neuropathy) and macrovascular (cardiovascular disease, stroke) complications as well as cognitive decline and gray matter atrophy in the brain, which could potentially increase the risk of dementia in patients with diabetes [49,50,51]. Maintaining stable blood glucose levels appears crucial to mitigate these risks. Our results showed a significant reduction in CV% in the treatment group but not in the control group. This confirms that the blood glucose became more stable when the dietary intervention was combined with the use of CGM.

We were not able to assess the effects of our intervention on HbA1c due to the 3-month average erythrocyte circulation time, but we were able to assess the GMI, which could give us estimated HbA1c values [52]. A significant reduction in GMI was observed in the treatment group but not in the control group, which potentially suggests a decrease in HbA1c and future diabetes-related complications [32,53]. However, we could not find any study to confirm the relationship between GMI and diabetes-related complications. Future studies should be conducted to clarify the relationship between GMI and HbA1c as well as how changes in GMI could relate to the risk of diabetes-related complications.

Our findings highlight that combining CGM with INT was both feasible and effective in improving blood glucose control in individuals with prediabetes who were overweight or obese, even in the absence of a weight loss diet. We previously showed that dietary compliance significantly increased in both groups over the course of the study, with a greater increase observed in the treatment group compared to the control group (*p* < 0.001) [25]. Additionally, both groups showed a numerical increase in time spent on physical activity throughout the study [25]. Therefore, this approach could offer an alternative method for lifestyle changes and better blood glucose management in individuals at high risk of chronic diseases like type 2 diabetes who struggle to achieve or maintain weight loss goals.

To our knowledge, this is the first study conducted to evaluate the effects of INT combined with real-time feedback from CGM in individuals with prediabetes and overweight or obesity. While there is a potential bias in dietary reporting when using 24-h dietary recalls, we aimed to minimize this by encouraging participants to promptly record their daily food and drink intake in a food diary. The treatment group was informed that their detailed dietary intake would be reviewed during each visit alongside CGM data to develop individualized dietary recommendations. The control group was informed that a dietary intake review would occur at the study’s end to give them recommendations about foods that are negatively affecting their blood glucose levels, ensuring consistent effort across both groups. We believe these methods provided participants with both a tool and an incentive to report their intake as accurately as possible. This study does have some limitations, including a small sample size and a short duration of follow-up. There was a lack of blinded CGM period for participants in the treatment group which could have affected their dietary intake before receiving nutrition education. Additionally, the effects of changes in various indicators of blood glucose control measured by CGM in individuals with prediabetes have not been extensively studied, which limits our ability to predict the potential effects of our intervention on the prevention or delay of type 2 diabetes and related complications. Further clinical trials with longer durations and larger populations are needed to confirm our findings.

## 5. Conclusions

Adding real-time feedback from CGM to individualized nutrition therapy positively affects blood glucose control in individuals with prediabetes who are overweight or obese. These findings suggest that incorporating CGM into personalized nutrition strategies could offer a more dynamic and effective approach to managing prediabetes. Clinically, this intervention may not only improve glycemic control but also empower at-risk individuals to make real-time adjustments to their diet, potentially preventing or delaying the progression to type 2 diabetes. Further research is needed to explore the long-term benefits and scalability of this approach in diverse populations.

## Figures and Tables

**Figure 1 nutrients-16-04005-f001:**
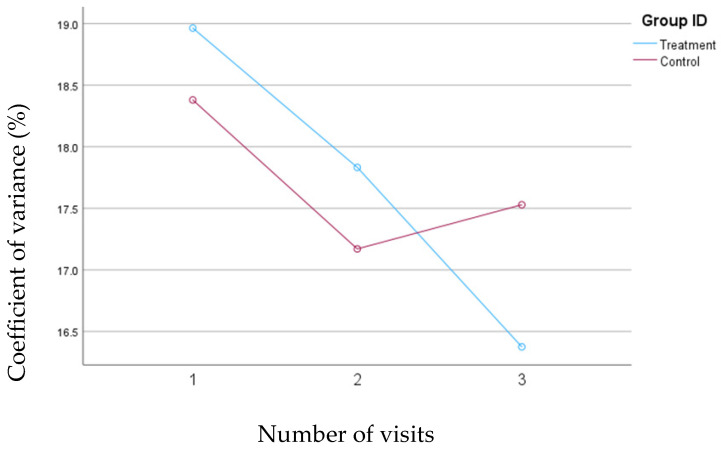
Mean changes in percent coefficient of variation (CV%) for blood glucose levels in the treatment and control groups during the 30-day follow-up period. The treatment group received individualized nutrition therapy and had real-time access to continuous glucose monitoring (CGM) results, while the control group was blinded to CGM results throughout this study.

**Figure 2 nutrients-16-04005-f002:**
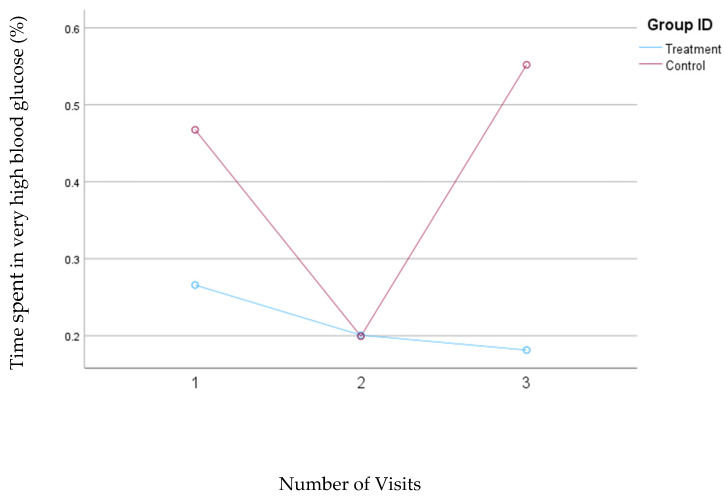
Mean changes in the percent of time spent in the very high blood glucose range in the treatment and control groups during the 30-day follow-up period. The treatment group received individualized nutrition therapy and had real-time access to continuous glucose monitoring (CGM) results, while the control group was blinded to CGM results throughout this study. Very high blood glucose was defined as a blood glucose concentration >250 mg/dL (Model 1).

**Figure 3 nutrients-16-04005-f003:**
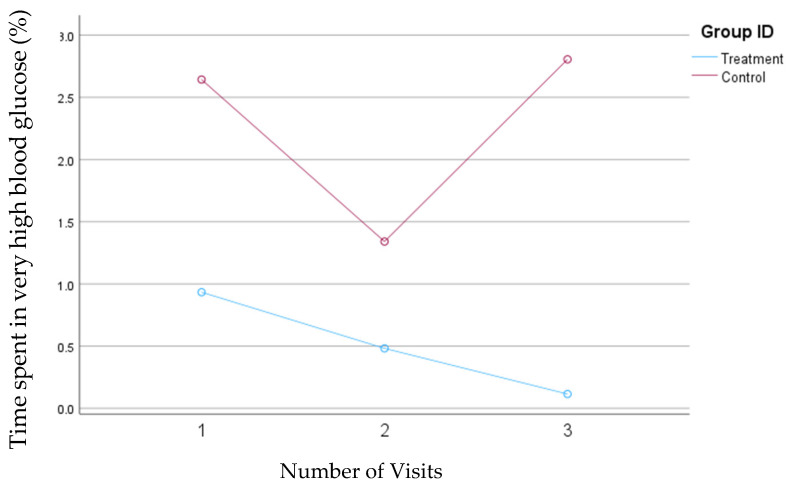
Mean changes in the percentage of time spent in the very high blood glucose range (>200 mg/dL) during the 30-day follow-up period. The treatment group, which received individualized nutrition therapy and had real-time access to continuous glucose monitoring (CGM), is compared to the control group, which received individualized nutrition therapy but was blinded to CGM results. Data are analyzed using Model 2.

**Figure 4 nutrients-16-04005-f004:**
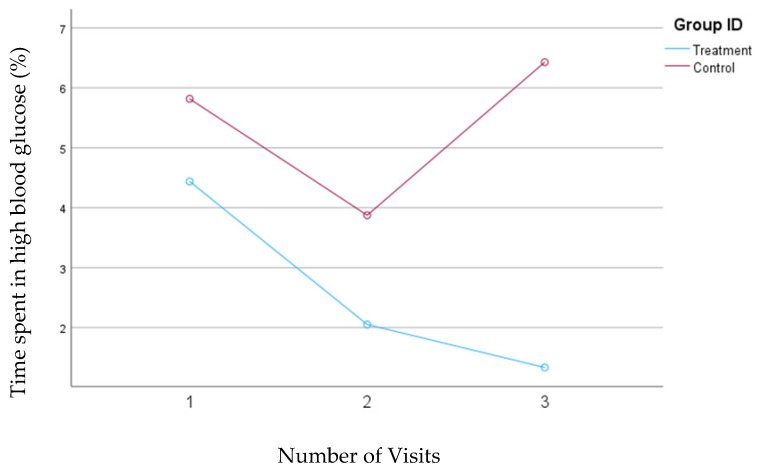
Mean changes in the percentage of time spent in the high blood glucose range (180–250 mg/dL) during the 30-day follow-up period. The treatment group, which received individualized nutrition therapy and had real-time access to continuous glucose monitoring (CGM), is compared to the control group, which received individualized nutrition therapy but was blinded to CGM results. Data are analyzed using Model 1.

**Figure 5 nutrients-16-04005-f005:**
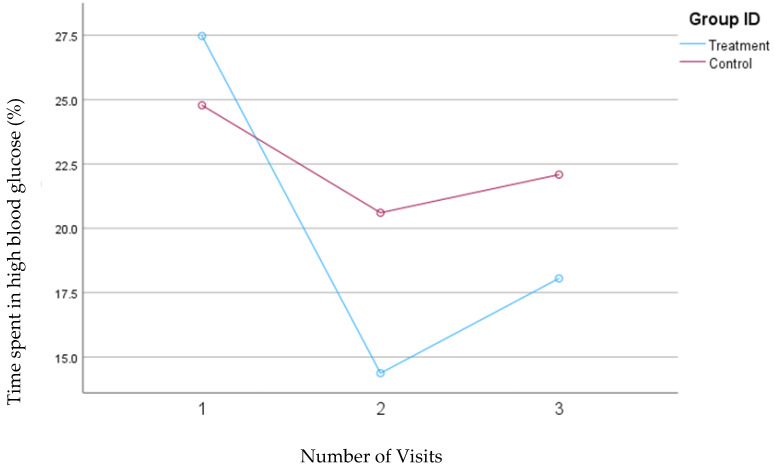
Mean changes in the percentage of time spent in the high blood glucose range (140–200 mg/dL, Model 2) during the 30-day follow-up period. The treatment group, which received individualized nutrition therapy and had real-time access to continuous glucose monitoring (CGM), is compared to the control group, which received individualized nutrition therapy but was blinded to CGM results.

**Figure 6 nutrients-16-04005-f006:**
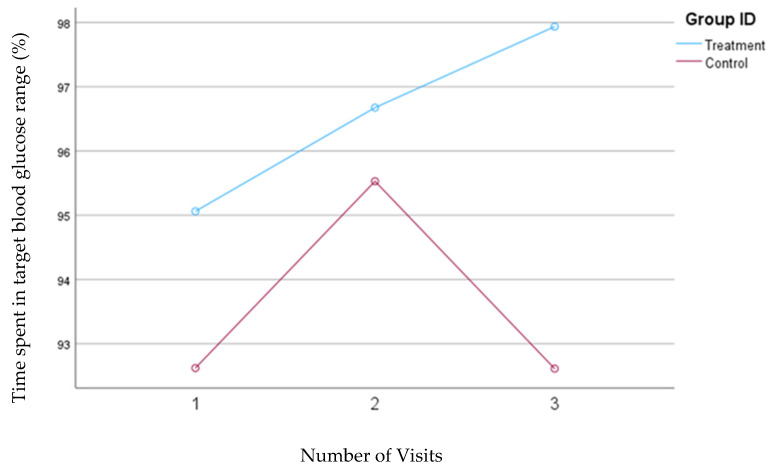
Mean changes in the percentage of time spent in the target range of glucose (70–180 mg/dL, Model 1) during the 30-day follow-up period. The treatment group, which received individualized nutrition therapy and had real-time access to continuous glucose monitoring (CGM), is compared to the control group, which received individualized nutrition therapy but was blinded to CGM results.

**Figure 7 nutrients-16-04005-f007:**
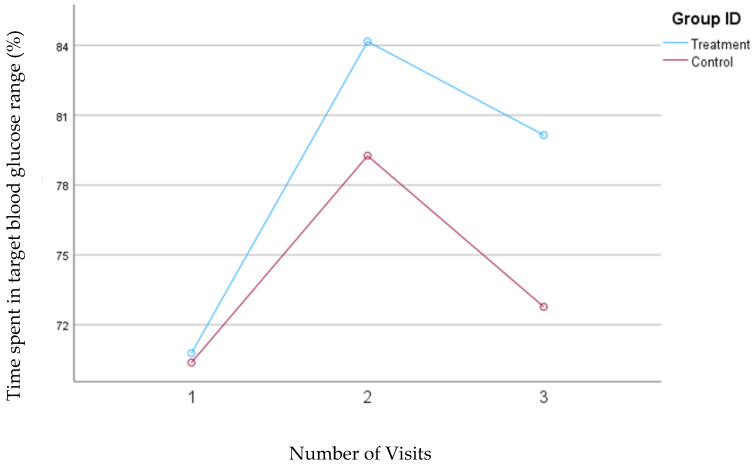
Mean changes in the percentage of time spent in the target blood glucose range (70–140 mg/dL, Model 2) during the 30-day follow-up period. The treatment group, which received individualized nutrition therapy and had real-time access to continuous glucose monitoring (CGM), is compared to the control group, which received individualized nutrition therapy but was blinded to CGM results.

**Table 1 nutrients-16-04005-t001:** Demographic characteristics of participants in each group.

Group	Treatment (*n* = 15)	Control (*n* = 15)	*p*-Value
Sex (female/male)	11/4	11/4	1.00
Age (Mean ± SD)	57.3 ± 5.2	52.7 ± 6.4	0.04
Race (White/Black/Asian)	9/1/5	10/2/3	
HbA1c%	5.7 ± 0.78	6.0 ± 0.2	0.26
BMI ^1^ (kg/m^2^)	31.8 ± 4.3	31.4 ± 4.5	0.84
Live alone (y/n)	1/14	2/13	0.55
Have medical insurance (y/n)	14/1	11/5	0.07
Employed (y/n)	11/4	12/3	0.7
Have financial concerns (y/n)	2/13	1/14	0.56
Visited RD ^2^ in the past	1/14	4/11	0.64

^1^ Body mass index. ^2^ Registered dietitian.

## Data Availability

The dataset is part of ongoing analyses. The raw data supporting the conclusions of this article will be made available by the authors upon request.

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
