# Peer review of "Enhancing the Impact of Individualized Nutrition Therapy with Real-Time Continuous Glucose Monitoring Feedback in Overweight and Obese Individuals with Prediabetes"

_nutrients, 2024, doi:10.3390/nu16234005_

Round 1

Reviewer 1 Report

Comments and Suggestions for Authors

In the manuscript, "Enhancing the Impact of Individualized Nutrition Therapy with Real-Time Continuous Glucose Monitoring Feedback in Overweight and Obese Individuals with Prediabetes" Basiri and Cheskin conducted a randomized study to evaluate the impact of personal CGM on glycemic parameters in adults with prediabetes. The manuscript is well written and investigates an important technology that may have significant implications for  diabetes prevention. The following concerns are raised however:

1) It is not completely clear the dietary intervention in the control group. Since both dietician and participant were blinded to the CGM, what was the specific intervention strategy in this group? 

2) For some of the glycemic metrics, it appears that there were significant differences at the first time interval when they were suppose to be on their usual diet? Analysis of the difference in diet is needed; however, it is possible that the introduction of the CGM altered the diet in those individuals in the treatment group. having a period of time when both were using blinded CGM would have been a useful control.

3) The authors mention that the participants tried different foods during this 3rd period of sensor wear. This should be analyzed regarding the dietary content to see if it in fact was associated with some of the increases in adverse glycemic outcomes.

4) The authors point out that the short duration is a limitation, but another significant limitation is the deviation from the diet and lack of blinded CGM as a baseline in the treatment group.

5) A significant missing analysis is the dietary recall. The authors should report on overall adherence to the macronutrient recommendations, changes in these overtime and between groups. 

Author Response

Dear Reviewer,

Thank you for taking the time to review our manuscript. We truly appreciate your valuable feedback. We have carefully considered and responded to all the comments, and we believe the revisions have strengthened the manuscript. Your insights have been instrumental in improving the quality of our work.

Please see below our response to your comments.

Thank you again for your thoughtful contributions.

1) It is not completely clear the dietary intervention in the control group. Since both dietician and participant were blinded to the CGM, what was the specific intervention strategy in this group?  

Thank you for requesting clarification. The dietary intervention for both groups was calculated based on their weight, height, gender, and age using the Mifflin-St Jeor equation. This information is provided in lines 118-121 of the manuscript and below:

“During this visit, all participants were given dietary recommendations tailored to their energy requirements for weight maintenance using the Mifflin St. Jeor formula [28], with a recommended macronutrient distribution of 50% carbohydrates, 20% protein, and 30% fat, constituting a moderate carbohydrate diet [29]”

2) For some of the glycemic metrics, it appears that there were significant differences at the first time interval when they were suppose to be on their usual diet? Analysis of the difference in diet is needed; however, it is possible that the introduction of the CGM altered the diet in those individuals in the treatment group. having a period of time when both were using blinded CGM would have been a useful control.

We agree with the reviewer's comments. We included the changes in the dietary intake of energy and macronutrients for both groups in lines 353-363. Unfortunately, we did not have a period of time when both were blinded to CGM results. This has been added to the limitation of the paper in lines 469-471. Please see below:

“During the study, the treatment group reduced their calorie intake from 1,774 kcal to 1,688 kcal, while the control group decreased from 2,148 kcal to 2,092 kcal. The percentage of calorie intake from carbohydrates declined from 40% to 36% in the treatment group and from 36% to 34% in the control group. The caloric intake from fat increased from 37% to 39% in the treatment group and from 39% to 40% in the control group. Similarly, the percentage of calories from protein rose from 20% to 22% in the treatment group and from 20% to 21% in the control group. All values have been adjusted for age.”

And

“There was a lack of blinded CGM period for participants in the treatment group which could have affected their dietary intake before receiving nutrition education.”

3) The authors mention that the participants tried different foods during this 3rd period of sensor wear. This should be analyzed regarding the dietary content to see if it in fact was associated with some of the increases in adverse glycemic outcomes.

Thank you for your comment. This information has been added to the manuscript. Please see lines 406-420 for further details.

“When analyzing the percentage of calorie intake from carbohydrates for the treatment group, no significant changes were observed from visit one (%39.68 ± 2.7) to visit two (%39.13 ± 2.38). However, a notable reduction was seen from visit two to visit three (%36.43 ± 2.52), confirming the positive effects of the intervention (INT) when combined with real-time feedback from CGM. Interestingly, no significant changes occurred between visit three and visit four (%36.17 ± 2.51), which aligns with anecdotal reports from participants who admitted to trying new food options during this period, anticipating the end of CGM access. In contrast, the control group showed a different pattern. Their carbohydrate intake decreased from visit one (%35.91 ± 2.67) to visit two (%33.79 ± 2.38) but remained relatively stable during visits three (%33.51 ± 2.52) and four (%33.98 ± 2.51). All reported values are adjusted for age. Longer-term studies may help address these effects by providing participants with sufficient time to adapt to dietary changes. Such an approach would allow participants to explore occasional foods of interest while maintaining more informed and sustainable dietary habits over time.”

4) The authors point out that the short duration is a limitation, but another significant limitation is the deviation from the diet and lack of blinded CGM as a baseline in the treatment group.

Thank you for your insightful comment. These limitations have been added to the limitations of the study in lines 469-471. Please see below.

There was a lack of blinded CGM period for participants in the treatment group which could have affected their dietary intake before receiving nutrition education.”

5) A significant missing analysis is the dietary recall. The authors should report on overall adherence to the macronutrient recommendations, changes in these overtime and between groups. 

This information has been added to the paper in lines 356-363. Please see below

“During the study, the treatment group reduced their calorie intake from 1,774 kcal to 1,688 kcal, while the control group decreased from 2,148 kcal to 2,092 kcal. The percentage of calorie intake from carbohydrates declined from approximately 40% to 36% in the treatment group and from 36% to 34% in the control group. The caloric intake from fat increased from around 37% to 39% in the treatment group and from 39% to 40% in the control group. Similarly, the percentage of calories from protein rose from about 20% to 22% in the treatment group and from 20% to 21% in the control group. None of these changes reached statistical significance, and all values were adjusted for age.”

Reviewer 2 Report

Comments and Suggestions for Authors

This is a well-designed randomized clinical trial evaluating the impact of integrating continuous glucose monitoring (CGM) with individualized nutrition therapy (INT) for blood glucose management in overweight or obese individuals with prediabetes. The authors concluded that the treatment group receiving real-time CGM feedback experienced significant improvements in blood glucose management metrics compared to the control group, who were blinded to their CGM data.

One of the issues that need to be addressed is potential bias in dietary reporting since self-reported food diaries may be inaccurate due to recall bias. How did the authors avoid this problem?
Also, assessing participants’ behavioral changes such as adherence to diet and physical activity should be one of the outcomes. One of the main benefits of CGM especially in non insulin users are life style changes. Have the authors considered using standardized questionnaires to address this isssues?

Author Response

Dear Reviewer,

Thank you for taking the time to review our manuscript. We truly appreciate your valuable feedback. We have carefully considered and responded to all the comments, and we believe the revisions have strengthened the manuscript. Your insights have been instrumental in improving the quality of our work.

Please see below our response to your comments.

Thank you again for your thoughtful contributions.

One of the issues that need to be addressed is potential bias in dietary reporting since self-reported food diaries may be inaccurate due to recall bias. How did the authors avoid this problem?

We acknowledge the reviewer’s concern regarding the limitations of dietary recalls. To mitigate these limitations, we implemented several strategies which are added to the manuscript lines 459-467. Please see below:

“While there is a potential bias in dietary reporting when using 24-hour dietary recalls, we aimed to minimize this by encouraging participants to promptly record their daily food and drink intake in a food diary and later enter it into the ASA24 system. The treatment group was informed that their detailed dietary intake would be reviewed during each visit alongside CGM data to develop individualized dietary recommendations. The control group was informed that a dietary intake review would occur at the study’s end to give them recommendations about foods that are negatively affecting their blood glucose levels, ensuring consistent effort across both groups. We believe these methods provided participants with both a tool and an incentive to report their intake as accurately as possible.”

Also, assessing participants’ behavioral changes such as adherence to diet and physical activity should be one of the outcomes. One of the main benefits of CGM especially in non insulin users are life style changes. Have the authors considered using standardized questionnaires to address this isssues?

Thank you for your comments. This information has been added to the manuscript lines 449-453.

We previously showed that dietary compliance significantly increased in both groups over the course of the study, with a greater increase observed in the treatment group compared to the control group (p<0.001) [25]. Additionally, both groups showed a numerical increase in time spent on physical activity throughout the study [25]”

Reviewer 3 Report

Comments and Suggestions for Authors

I read with interest the manuscript “Enhancing the Impact of Individualized Nutrition Therapy 2 with Real-Time Continuous Glucose Monitoring Feedback in 3 Overweight and Obese Individuals with Prediabetes” by Basiri et al.

English is fine.

1.       Introduction:
- While the introduction effectively emphasizes the high prevalence of prediabetes, it would benefit from a more explicit discussion of how CGM has transformed real-time feedback in clinical interventions. This could further reinforce the study’s relevance.

-            The studies cited regarding the short-term efficacy of weight loss interventions (lines 54–57) are pertinent, but the introduction lacks a clear transition to how CGM feedback could specifically address the psychological and adherence challenges mentioned.Materials and Methods:

2.       Methods
- Study Population: The inclusion and exclusion criteria are comprehensive, but the rationale for age limitation (45-65 years) is not provided. Explaining why this age group was chosen would help readers assess the generalizability of findings.

-            Randomization and Blinding: It’s unclear how participants were randomized into treatment and control groups. A more detailed explanation of the randomization process and any potential blinding methods would improve the study’s transparency.

-            Dietary Recommendations: The authors provide an outline of dietary recommendations, but additional details on how adherence to these recommendations was monitored and how deviations were handled would enhance understanding of intervention fidelity.

-            Statistical Analysis: The statistical approach is generally appropriate for the study design, but additional clarification on how potential confounders were identified and adjusted for during analysis would be beneficial. Also, a discussion of the statistical power of the study, given the sample size (n=30), would strengthen confidence in the reliability of the findings.

3.       Discussion

-            Clinical Implications of CGM Integration: Emphasize that using CGM to provide real-time data and feedback is key to sustaining dietary intervention effects on glycemic control. Highlight the potential for CGM to enhance INT outcomes, particularly with regard to reducing fluctuations in blood glucose levels, a significant risk factor for complications.

-            Importance of TIR: Discuss the significance of TIR improvements and relate this to the risk reduction for diabetes-related complications, even during prediabetes.

-            Potential Confounders: The increase in high glucose ranges near study completion in the treatment group (possibly due to dietary exploration by participants) is an interesting insight. Suggest how longer-term studies or additional dietary guidance may mitigate such effects.

-            Recommendation for Extended Follow-up: Considering your short study duration, propose that future studies should examine the sustainability of dietary and CGM interventions over a longer period.

Author Response

Dear Reviewer,

Thank you for taking the time to review our manuscript. We truly appreciate your valuable feedback. We have carefully considered and responded to all the comments, and we believe the revisions have strengthened the manuscript. Your insights have been instrumental in improving the quality of our work.

Please see below our response to your comments.

Thank you again for your thoughtful contributions.

  1. Introduction:
    - While the introduction effectively emphasizes the high prevalence of prediabetes, it would benefit from a more explicit discussion of how CGM has transformed real-time feedback in clinical interventions. This could further reinforce the study’s relevance.

-            The studies cited regarding the short-term efficacy of weight loss interventions (lines 54–57) are pertinent, but the introduction lacks a clear transition to how CGM feedback could specifically address the psychological and adherence challenges mentioned.

Thank you for your comment. A new citation was added to the manuscript lines 75 to 82 regarding the change in lifestyle and adherence to dietary recommendations using CGM. Please also see below:

“It has been shown that positive psychological reactions, including reduced stress and anxiety, greater peace of mind, and a greater sense of normalcy, can be achieved through effective CGM use[23]. A randomized controlled trial by Anh et al. showed that non-contact dietary coaching, combined with CGM, improved behavioral skills and health outcomes in adults with prediabetes or diabetes [24]. Moreover, it has been shown that a personalized dietary intervention approach combined with CGM can promote better adherence to dietary recommendations [25].”

  1. Methods
    - Study Population: The inclusion and exclusion criteria are comprehensive, but the rationale for age limitation (45-65 years) is not provided. Explaining why this age group was chosen would help readers assess the generalizability of findings.

The rationale for choosing the age range has been added to the manuscript lines 96-97. Please see below:

“The rationale for the age range is that the prevalence of prediabetes is higher in this age group [27,28]

-            Randomization and Blinding: It’s unclear how participants were randomized into treatment and control groups. A more detailed explanation of the randomization process and any potential blinding methods would improve the study’s transparency.

Thank you for asking for clarification. This information has been added to the manuscript lines 111 and 134-136. Please see below:

“Participants were then randomized into treatment (n=15) or control groups (n=15) using an online random number generator.”

And

“In contrast, both the dietitian and control group participants were blinded to the CGM recordings until the end of the study; thus, the CGM data was not included in nutrition education for the control group.”

-            Dietary Recommendations: The authors provide an outline of dietary recommendations, but additional details on how adherence to these recommendations was monitored and how deviations were handled would enhance understanding of intervention fidelity.

This information has been added to the manuscript lines 137-141 and also below:

“At visits three and four, participants were also asked how well they felt they were able to follow the provided recommendations. If participants reported any obstacles in adherence, the dietitian collaborated with them to address these issues and identify practical alternatives, enhancing the likelihood of successful adherence to the recommendations.”

-            Statistical Analysis: The statistical approach is generally appropriate for the study design, but additional clarification on how potential confounders were identified and adjusted for during analysis would be beneficial. Also, a discussion of the statistical power of the study, given the sample size (n=30), would strengthen confidence in the reliability of the findings.

The information about potential confounders and how they were handled is provided in lines 205-209 and 216-222. Details about calculating power and sample size have been added to the manuscript lines 195-202. Please see below:

“Descriptive statistics were performed to evaluate population characteristics, and an Analysis of Variance (ANOVA) table was used to assess the distribution of covariates between the groups. If there was a significant difference in the distribution of a potential confounding variable between the groups, the variable was included as a covariate in the model to minimize its potential effects on outcome measures”

And

“The distribution of potential covariates was evaluated, and no significant differences were observed for sex, ethnicity, BMI, and hemoglobin A1c (HbA1c) between groups; however, the average age of participants was significantly higher in the treatment group compared to the control group. Therefore, age was included as a covariate in the models. Given the established relationship between BMI and blood glucose control, BMI was also included as a covariate in the models to account for its potential confounding effects. “

And  

“A power analysis was performed to determine the required sample size for detecting a statistically significant interaction between the intervention (INT versus control) and time (across four time points). The calculation was based on an effect size of 0.25, representing the mean difference and standard deviation for the primary outcome of interest: a clinically meaningful reduction in glucose variability over 30 days [37]. Using G*Power software (version 3.1.9.4), we estimated that a sample of at least 24 participants (12 per group) would be needed to achieve 80% power at a 0.05 significance level. To allow for an anticipated 20% dropout or missing data rate, we ultimately enrolled 30 participants.”

  1. Discussion

-            Clinical Implications of CGM Integration: Emphasize that using CGM to provide real-time data and feedback is key to sustaining dietary intervention effects on glycemic control. Highlight the potential for CGM to enhance INT outcomes, particularly with regard to reducing fluctuations in blood glucose levels, a significant risk factor for complications.

Thank you very much for the suggestion. This information has been added to the manuscript lines 447- 452 and 430-437. Please see below:

“Our findings highlight that combining CGM with INT was both feasible and effective in improving blood glucose control in individuals with prediabetes who were overweight or obese, even in the absence of a weight-loss diet. We previously showed that dietary compliance significantly increased in both groups over the course of the study, with a greater increase observed in the treatment group compared to the control group (p<0.001)[25].”

And

 “Fluctuations in blood glucose levels are an important risk factor for the development and progression of various microvascular (nephropathy, retinopathy, neuropathy) and macrovascular (cardiovascular disease, stroke) complications as well as cognitive decline and gray matter atrophy in the brain, which could potentially increase the risk of dementia in patients with diabetes [49–51]. Maintaining stable blood glucose levels appears crucial to mitigate these risks. Our results showed a significant reduction in CV% in the treatment group, but not in the control group. This confirms that the blood glucose became more stable when the dietary intervention was combined with the use of CGM. ”

-            Importance of TIR: Discuss the significance of TIR improvements and relate this to the risk reduction for diabetes-related complications, even during prediabetes.

Thank you for the suggestion. This information can be found in the manuscript lines 375-379 and lines 422-428. Please see below:

“It has been found that TIR is a strong predictive measure for long-term complications of diabetes, particularly microvascular complications [39]. Furthermore, reduced time at high and very high blood glucose levels in patients with diabetes is related to reduced albuminuria, severity of diabetic retinopathy, and prevalence of peripheral neuropathy and cardiac autonomic neuropathy [40].”

 and

“It is well documented that microvascular complications like retinopathy, neuropathy, and nephropathy, as well as macrovascular complications like cardiovascular disease and kidney disease, can manifest during the prediabetes stage, before progression to overt diabetes [46–48]. Therefore, the observed reductions in time spent in the very high and high blood glucose ranges and significant improvements in TIR, CV, and blood glucose concentrations in the treatment group suggest a significant potential for preventing the onset of related complications in this population.”

-            Potential Confounders: The increase in high glucose ranges near study completion in the treatment group (possibly due to dietary exploration by participants) is an interesting insight. Suggest how longer-term studies or additional dietary guidance may mitigate such effects.

Thank you for your great suggestion. This information has been included in the paper lines 416-420. Please see below:

“Longer-term studies may help address these effects by providing participants with sufficient time to adapt to dietary changes. Such an approach would allow participants to explore occasional foods of interest while maintaining more informed and sustainable dietary habits over time.”

-            Recommendation for Extended Follow-up: Considering your short study duration, propose that future studies should examine the sustainability of dietary and CGM interventions over a longer period.

Thank you for your comment. This information can be found in lines 428-429. Please see below:

“Longer-duration clinical trials should be conducted to assess the sustainability of similar interventions.”

Reviewer 4 Report

Comments and Suggestions for Authors

The manuscript submitted for publication to Nutrients by Basiri and Cheskin titled: "Enhancing the Impact of Individualized Nutrition Therapy with Real-Time Continuous Glucose Monitoring Feedback in Overweight and Obese Individuals with Prediabetes" is an interesting human study aiming to investigate the relationship between continuous glucose monitoring and the impact of personalized nutrition therapy in people with prediabetes in the overweight and obese status. 

The topic investigated is interesting with clinical implications and significant potential for application and possibly improvement of strategies regarding individualized management to reduce risk of diabetes. 

The manuscript is well written and structured with a good organization and flow. 

The reviewer would like to offer the following points below for the authors' consideration: 

1. In addition to the US statistics provided in the introduction section, consider adding some brief overview statistics at the global level from WHO and/or International Diabetes Federation. Also, it would be interesting to include briefly the cost/burden of disease.

2. BMI is an index and is thus calculated not measured (so it does not really have units). Although it is not uncommonly reported with units this is actually wrong.

3. Consider providing the rationale for the sample size selection.

4. The authors mentioned in the Methods section that taking appetite suppressants is an exclusion criterion (and rightfully so). Did they also consider other oral hypoglycemics (such as low dose metformin medication not atypically prescribed to people with pre-diabetes)? 

5. Do the authors consider physical activity and meal timing in their analyses of the obtained data? 

6. Given the small sample size consider designating the study as a pilot.

7. Consider expanding the conclusions section a bit and potentially adding to the clinical significance the findings may extend. 

Good job overall.

Author Response

Dear Reviewer,

Thank you for taking the time to review our manuscript. We truly appreciate your valuable feedback. We have carefully considered and responded to all the comments, and we believe the revisions have strengthened the manuscript. Your insights have been instrumental in improving the quality of our work.

Please see below our response to your comments.

Thank you again for your thoughtful contributions.

  1. In addition to the US statistics provided in the introduction section, consider adding some brief overview statistics at the global level from WHO and/or International Diabetes Federation. Also, it would be interesting to include briefly the cost/burden of disease.

Thank you for your insightful comment. The requested information has been added to the manuscript lines 44 to 51. Please see below:

“Globally, 537 million adults aged 20 to 79—about 1 in 10—are currently living with diabetes [5]. This number is expected to increase significantly, reaching 643 million by 2030 and 783 million by 2045. In 2021 alone, diabetes was responsible for 6.7 million deaths, equating to one death every five seconds [5]. The financial burden is also substantial, with diabetes-related health expenditure reaching at least USD 966 billion, a 316% increase over the past 15 years [5]. Additionally, 541 million adults have Impaired Glucose Tolerance (IGT), putting them at a high risk of developing type 2 diabetes [5]”

  1. BMI is an index and is thus calculated not measured (so it does not really have units). Although it is not uncommonly reported with units this is actually wrong.

Thank you for pointing out that BMI is a calculated index and technically does not have units. However, to maintain consistency with common reporting practices in the field and align with other publications, we prefer to report BMI with units (kg/m²) in this manuscript.

  1. Consider providing the rationale for the sample size selection.

Thank you for your comment. The rationale for the sample size and power calculation has been added to the manuscript lines 195-202.

“A power analysis was performed to determine the required sample size for detecting a statistically significant interaction between the intervention (INT versus control) and time (across four time points). The calculation was based on an effect size of 0.25, representing the mean difference and standard deviation for the primary outcome of interest: a clinically meaningful reduction in glucose variability over 30 days [37]. Using G*Power software (version 3.1.9.4), we estimated that a sample of at least 24 participants (12 per group) would be needed to achieve 80% power at a 0.05 significance level. To allow for an anticipated 20% dropout or missing data rate, we ultimately enrolled 30 participants.”

  1. The authors mentioned in the Methods section that taking appetite suppressants is an exclusion criterion (and rightfully so). Did they also consider other oral hypoglycemics (such as low dose metformin medication not atypically prescribed to people with pre-diabetes)? 

Thank you for your comment. Only one participant in the control group had been taking metformin 500mg/day for more than 3 months. This information has been added to the manuscript lines 214-216. Please see below:

“Except for one participant in the control group who had been taking metformin 500 mg/day for over three months, the participants were not using any blood glucose-lowering medications.”

  1. Do the authors consider physical activity and meal timing in their analyses of the obtained data? 

Thank you for your comment. We have added more details about participants' dietary intake and physical activity in lines 355-363 and 449-453 of the manuscript. Unfortunately, we were unable to analyze meal timing. Please see the updated information below.

“During the study, the treatment group reduced their calorie intake from 1,774 kcal to 1,688 kcal, while the control group decreased from 2,148 kcal to 2,092 kcal. The percentage of calorie intake from carbohydrates declined from approximately 40% to 36% in the treatment group and from 36% to 34% in the control group. The caloric intake from fat increased from around 37% to 39% in the treatment group and from 39% to 40% in the control group. Similarly, the percentage of calories from protein rose from about 20% to 22% in the treatment group and from 20% to 21% in the control group. None of these changes reached statistical significance, and all values were adjusted for age.”

And

“We previously showed that dietary compliance significantly increased in both groups over the course of the study, with a greater increase observed in the treatment group compared to the control group (p<0.001) [25]. Additionally, both groups showed a numerical increase in time spent on physical activity throughout the study [25]”

  1. Given the small sample size consider designating the study as a pilot.

Thank you for your suggestion. While we understand that smaller studies are sometimes classified as pilot studies, we conducted a power analysis to ensure that our sample size of 30 participants was sufficient to detect a statistically significant effect with 80% power at a 0.05 significance level. Based on these calculations, we believe the sample size was adequate to meet the objectives of the study and provide meaningful results. This information now has been added to the manuscript lines 195-202. Please see below:

“A power analysis was performed to determine the required sample size for detecting a statistically significant interaction between the intervention (INT versus control) and time (across four time points). The calculation was based on an effect size of 0.25, representing the mean difference and standard deviation for the primary outcome of interest: a clinically meaningful reduction in glucose variability over 30 days[37]. Using G*Power software (version 3.1.9.4), we estimated that a sample of at least 24 participants (12 per group) would be needed to achieve 80% power at a 0.05 significance level. To allow for an anticipated 20% dropout or missing data rate, we ultimately enrolled 30 participants.”

  1. Consider expanding the conclusions section a bit and potentially adding to the clinical significance the findings may extend. 

Thank you for your insightful comment. We have expanded the conclusions section in response to your recommendation. Please see lines 479-485 and the paragraph below for further details.

 “These findings suggest that incorporating CGM into personalized nutrition strategies could offer a more dynamic and effective approach to managing prediabetes. Clinically, this intervention may not only improve glycemic control but also empower at-risk individuals to make real-time adjustments to their diet, potentially preventing or delaying the progression to type 2 diabetes. Further research is needed to explore the long-term benefits and scalability of this approach in diverse populations.”

Round 2

Reviewer 3 Report

Comments and Suggestions for Authors

I am satisfied with author changes